# The interplay of Rac1 activity, ubiquitination and GDI binding and its consequences for endothelial cell spreading

Jisca Majolée[1], Fabienne Podieh[1], Peter L. Hordijk[1], Igor Kovačević[1,2]*

**1** Department of Physiology, Amsterdam University Medical Centers, Amsterdam, The Netherlands,
**2** Department of Gene Regulation, Institute of Physiological Chemistry, Martin-Luther University Halle-Wittenberg, Halle (Saale), Germany

☯ These authors contributed equally to this work.
* igor.kovacevic@uk-halle.de

**Data Availability Statement:** All relevant data are within the manuscript and its S1 Raw images and S1–S4 Figs.

## Abstract

Signaling by the Rho GTPase Rac1 is key to the regulation of cytoskeletal dynamics, cell spreading and adhesion. It is widely accepted that the inactive form of Rac1 is bound by Rho GDI, which prevents Rac1 activation and Rac1-effector interactions. In addition, GDI-bound Rac1 is protected from proteasomal degradation, in line with data showing that Rac1 ubiquitination occurs exclusively when Rac1 is activated. We set out to investigate how Rac1 activity, GDI binding and ubiquitination are linked. We introduced single amino acid mutations in Rac1 which differentially altered Rac1 activity, and compared whether the level of Rac1 activity relates to Rac1 ubiquitination and GDI binding. Results show that Rac1 ubiquitination and the active Rac1 morphology is proportionally increased with Rac1 activity. Similarly, we introduced lysine-to-arginine mutations in constitutively active Rac1 to inhibit site-specific ubiquitination and analyze this effect on Rac1 signaling output and ubiquitination. These data show that the K16R mutation inhibits GTP binding, and consequently Rac1 activation, signaling and–ubiquitination, while the K147R mutation does not block Rac1 signaling, but does inhibits its ubiquitination. In both sets of mutants, no direct correlation was observed between GDI binding and Rac1 activity or -ubiquitination. Taken together, our data show that a strong, positive correlation exists between Rac1 activity and its level of ubiquitination, but also that GDI dissociation does not predispose Rac1 to ubiquitination.

## Introduction

Regulation of the actin cytoskeleton underlies essential aspects of eukaryotic cell behavior, including supporting cell structure and enabling directional migration. The main orchestrators of cytoskeletal rearrangements are the Rho GTPases, each with distinctive regulatory features. Signaling by the small Rho GTPase Rac1 typically induces cell spreading, whereas activity of its closely related siblings RhoA and RhoB induces stress fiber formation and cell contraction [1, 2]. Rho GTPases cycle between a GDP-bound 'off' state and a GTP-bound 'on' state. When GTP-bound, the conformation of Rho GTPases allows for their interaction with

**Funding:** FP was funded by NWO grant OCENW.
klein.021. The funders had no role in study design,
data collection and analysis, decision to publish, or
preparation of the manuscript.

**Competing interests:** The authors have declared
that no competing interests exist.

downstream effectors. The exchange of GDP for GTP, activating the GTPase, is catalyzed by Guanine nucleotide Exchange Factors (GEFs). The intrinsic GTPase activity of Rho GTPases, driving the hydrolysis of GTP to GDP, is accelerated by so-called GTPase-activating Proteins (GAPs) [3].

In a resting cell, the majority of the Rac1 GTPase is inactive. In this inactive state, Rac1 is bound to a Guanine nucleotide Dissociation Inhibitor (GDI) which prevents both its interaction with the plasma membrane and its activation by GEFs [4, 5]. Rho GTPases require to be associated to a membrane to release the RhoGDI and undergo GEF-mediated activation [5]. Whereas most models of the Rac1 activation cycle suggest that active Rac1 is not bound by the GDI, several papers have shown an interaction between a constitutively active Rac1 (G12V) and the GDI and a lack of GDI binding by the constitutively inactive mutant of Rac1 (T17N) [6–8].

In addition to its regulation by GEFs, GAPs and the GDI, post-translational modification (PTMs) of Rac1 has gained increasing interest, providing additional insights in its regulation [9]. Rac1, similar to many other small GTPases, is modified by C-terminal lipidation, which is involved in both GDI binding as well as Rac1-membrane association [7]. In addition, Rac1 is subject to post-translational modification by the covalent attachment of ubiquitin or ubiquitin-like proteins [10].

Ubiquitination has, so far, mainly been implicated in the proteasome-mediated degradation of the active, but not the inactive, form of Rac1 [11–13]. Several ubiquitin E3 ligases have been linked to the regulation of Rac1. Members of the inhibitor of apoptosis protein family (IAPs) [13] and the E3 HECT ligase HACE1 ubiquitinate active Rac1 at K147, targeting the protein for proteasomal degradation [14, 15]. In contrast, however, the E3 ligase TRAF6 was suggested to activate Rac1 via K63-linked poly-ubiquitination at Rac1 K16 [16, 17].

Both the use of Rac1-activating stimuli (e.g., growth factors or bacterial toxins) and of 'constitutively active' mutants have facilitated studies on Rac1 signaling [18]. The constitutively active Rac1 Q61L mutant, which is incapable of GTP hydrolysis, has been used widely in fundamental studies on the cellular consequences of Rac1 signaling [19–21]. However, experiments with this Rac1 mutant may give an incomplete view, as GTP/GDP cycling is an important aspect of Rho GTPase regulation and signaling [22]. Another, commonly used, dominant active version of Rac1 has a G12V mutation, in which GTP hydrolysis activity is 6-fold decreased as compared to Rac1 WT [22].

In addition to these well-known Rac1 Q61L and G12V mutations, several other 'activating' mutations of Rac1 have been identified, a.o. in cancer patients. The Rac1 F28L and P29S mutations have been extensively analyzed *in vitro*. These mutations induce rapid GDP dissociation and fast GDP-GTP cycling, ultimately increasing the activity of Rac1 [23–25]. Tyrosine phosphorylation of Rac1 at Y64 has been linked to its inactivation. Mutation of Y64 to a non-phosphorylatable phenylalanine (Y64F) leads to decreased GDI binding, and has been proposed to increase Rac1-GTP levels [26]. Similarly, mutation of R66 inhibits the binding of GDI to Rac1, theoretically exposing it to activation and/or degradation [26]. The oncogenic, activating Rac1 N92I mutation was discovered in the human sarcoma cell line HT1080 [20] and the C157Y mutation was discovered in patients with cranial malformations and has, as yet, unclear effects on Rac1 activity [20, 27]. All these single amino acid mutations affect Rac1 activity by different, but in some cases poorly defined, means. Analysis of these Rac1 mutants may provide new insights in Rac1 regulation and signaling.

In this study, we analyzed two sets of Rac1 mutants: (i) a series of Rac1 mutants encoding different activating mutations, and (ii) a set of lysine-to-arginine (K-R) mutants in activated Rac1 to prevent its ubiquitination. We expressed these mutant proteins and the appropriate controls in human endothelial cells (ECs) and HEK293T cells, and analyzed (i) their

localization, activity and ubiquitination, and (ii) effects of the mutations on GDI binding and induction of endothelial cell spreading. Our data reveal a strong correlation between the level of Rac1 activity and (i) its level of ubiquitination and (ii) its morphological effects in ECs. RhoGDI binding, however, does not correlate with the ubiquitination- or morphological changes induced by activated Rac1 mutants. These data show that the activity of Rac1, rather than its interaction with RhoGDI, determines its signaling capacity and susceptibility to ubiquitination.

## Materials and methods

### Antibodies and reagents

Antibodies used in this study were anti-Myc-tag (#2278), anti-GAPDH (#2118), anti-RhoGDI (#2564), all from Cell Signaling, anti-Rac1 (#610650, BD Transduction Laboratories), anti-mCherry (#NBP2-25157, Novus Bio) and anti-HA-tag (#H3663, Sigma)

For immunofluorescent staining, myc-Rac1 was stained using anti-Myc-tag (#2278, Cell Signaling) as primary antibody in combination with the secondary antibody Alexa 488 donkey anti-rabbit (#A32790, Invitrogen). In all stainings, nucleus was stained with DAPI (#62248, Thermo Fisher scientific) and F-actin with Acti-stain 670 phalloidin (#PHDN1-A, Cytoskeleton).

The inhibitors PR619 (#S7130) and MLN7243 (#S8341) were used at 2.5 µM concentration. MG132 (#S2619) was used at 5 µM concentration. All were from Selleck Chemicals.

### Cell culture

**HUVECs.** Primary HUVECs were purchased from Lonza (#CC-2519) and cultured on fibronectin (5 µg/ml)-coated plates in Endothelial Cell medium (ECM) (ScienCell Research Laboratories). Cells were cultured at 37°C and 5% $CO_2$ and the medium was refreshed every second day. Experiments were performed with cells until passage 5.

**HEK293T cells.** HEK293T cells (ATCC) (#CRL-3216) were grown in Dulbecco's Modified Eagle Medium (Gibco) (#41966–029) supplemented with penicillin 100 U/mL and streptomycin 100 µg/mL, L-glutamine 2 mMol/L (all from Bio Whittaker/Lonza), 1 mM sodium pyruvate (Gibco) (#11360–070) and 10% Fetal Bovine Serum (PAA) (#A15-101).

### Mutagenesis

Single mutants of mCherry-Rac1 were generated using the site-directed mutagenesis. Mutations were introduced into the pmCherryC1-Rac1 vector in a PCR reaction using site-specific primers (Invitrogen) and high-fidelity Phusion DNA polymerase (NEB). Template DNA was digested by DpnI (NEB), and PCR product was transformed into competent DH5α Escherichia coli (NEB). Bacterial colonies were screened for presence of the desired mutation by DNA sequencing.

### Plasmid expression

**HEK293T cells.** Plasmid overexpression in HEK293T cells was done using TransIT-LT1 (MirusBio) according to the manufacturer's protocol.

**HUVEC.** Ectopic expression in HUVECs was done using the 4D-Nucleofector™ system (Lonza). Per plasmid, 10 cm² subconfluent HUVECs were electroporated with 2 µg of DNA with transfection protocol CA-167.

## Western blot

Cells were washed with PBS supplemented with 1mM $CaCl_2$ and 0.5 mM $MgCl_2$ and lysed in 2x SDS sample buffer (125 mM Tris-HCl pH 6.8, 4% SDS, 20% glycerol, 100 mM DTT, 0.02% Bromophenol Blue in MilliQ). Proteins were separated by SDS-PAGE and transferred to nitro-cellulose membranes, followed by incubation with designated primary antibodies in 5% BSA in TBS-T. Protein bands were visualized with enhanced chemiluminescence (Amersham/GE-healthcare) on the AI600 machine (Amersham/GE-healthcare).

## Immunofluorescence staining and confocal imaging

HUVECs were seeded on fibronectin-coated (5 μg/ml) 2 $cm^2$ coverslips (Thermo Scientific, Menzel-gläser) (#10319303). Warm (37˚C) 4% paraformaldehyde (Sigma Aldrich) (#158127) in phosphate buffered saline (PBS) (B Braun) (#3623140) was added onto the cells and incubated at room temperature for 15 minutes. After three washes with PBS, cells were permeabilized with 0,2% triton X-100 in PBS for 3 minutes and blocked for 30 minutes with 1% HSA in PBS. Coverslips were stained with primary antibodies in 1% HSA/PBS for 1 hour at room temperature or overnight at 4˚C. After washing with PBS, coverslips were incubated with a FITC-labeled secondary antibody (anti-rabbit or anti-mouse 1:100 in 1% HSA/PBS), Acti-stain 670 phalloidin (Cytoskeleton) (#PHDN1-A) and DAPI (Thermo Fisher Scientific) for 1 hour at room temperature. Coverslips were mounted on Mowiol4-88/DABCO solution (Calbiochem, Sigma Aldrich). Confocal scanning laser microscopy was performed on a Nikon A1R confocal microscope (Nikon). Images were analyzed and equally adjusted with ImageJ software.

## CRIB pulldown

To analyze Rac activity, a 6-cm dish (21 $cm^2$) of HUVECs or 10 $cm^2$ of HEK293T were washed once with PBS supplemented with 1 mM $CaCl_2$ and 0.5 mM $MgCl_2$ and lysed in 500 μl of cold Lysis buffer (150 mM NaCl, 50 mM Tris pH 7.6, 1% Triton-X 100, 20 mM $MgCl_2$) with 30 ug CRIB peptide. Cell debris was spun down at 14.000 rpm for 5 minutes, after which 10% of lysate was mixed with 3x SB (187.5 mM Tris-HCl pH 6.8, 6% SDS, 30% glycerol, 150 mM DTT, 0.03% Bromophenol Blue in MilliQ) (input) and the remaining lysate was incubated with Streptavidin beads for 30 minutes rotating at 4˚C. Next, the beads were washed 5 times in lysis buffer with freshly added 10mM $MgCl_2$, all buffer was aspirated and the beads were lysed in 30–50 μl 2x SB (pulldown). Rac1 in the input and pulldown samples was analyzed by Western Blot.

## Rac1 ubiquitination assay in cells

pcDNA3-HA-Ubiquitin was co-transfected with pmCherryC1-Rac1 or pcDNA-2x myc-Rac1 mutants into HEK293T cells using TransIT (Mirus) according to the manufacturer's protocol. Before lysis, cells were treated with 5 μM MG132 for 4 hrs. For analysis of ubiquitination of Rac1, denaturing HA-immunoprecipitation was performed at 24 h after transfection as previously described [28].

## Co-IP of myc- and mCherry-tagged Rac1

Myc- or mCherry-tagged Rac1 was transiently expressed in HEK293T cells. After 16 hours, Co-IP was performed using the Myc-TRAP or RFP-TRAP kit (Chromotek) according to the manufacturer's protocol.

## Results

### Rac1 activity is regulated by the ubiquitin-proteasome system

Our group previously identified Cullin-3 as the specific ubiquitin E3 ligase for the GTPase RhoB, and showed that decreased levels of Cullin-3 activity induced a rapid increase in RhoB protein level and activity with consequent, F-actin mediated contraction in endothelial cells [29]. Since active Rac1 is prone to ubiquitination, we tested whether the application of general inhibitors of the ubiquitin-proteasome system (UPS) would influence Rac1 activity in ECs. We treated confluent HUVEC monolayers with MG132 (proteasome inhibitor), MLN7243 (E1 ubiquitin activating enzyme inhibitor), PR619 (Deubiquitinating (DUB) enzyme inhibitor) or a combination of PR169 and MG132 and measured Rac1 activity by CRIB pulldown (Fig 1A and 1B). Total levels of Rac1 were unchanged, but we observed a clear increase in Rac1 activity induced by MLN7243, and a decrease in Rac1 activity induced by PR619. Inhibition of proteasomal degradation by MG132 slightly elevated Rac1 activity levels, but simultaneous treatment of PR619 and MG132 was similar to PR619 treatment only. The inhibitors used do not directly interfere with the binding of Rac1 to the PAK CRIB domain (S1 Fig). These data show that short-term inhibition of the cellular machinery that controls (de-)ubiquitination has differential effects on Rac1 activity in ECs, without significantly affecting Rac1 levels. This confirms that, in addition to GEF- and GAP-mediated regulation of Rac1 activity, direct or indirect modification by ubiquitin plays an important role in the induction and/or stability of its GTP-bound, active form.

### Rac1 activity levels correlate positively with Rac1 ubiquitination

After analyzing the effect of inhibitors of the UPS system on Rac1 activity, we set out to investigate whether differential Rac1 activity levels could be regulating its ubiquitination. We introduced single amino acid mutations in mCherry-Rac1 that were previously found to increase Rac1 activity (Fig 2A and 2B) and compared their activity by performing CRIB pulldown assays after expression in HEK293T cells (Fig 2C). As supported by published literature, we found different levels of mCherry-Rac1 activation in the different mutants when compared to WT mCherry-Rac1. Rac1 G12V and Q61L, included as positive controls, showed an average 9.0- and 11.2-fold increase in Rac1 activity compared to Rac1 WT, respectively, followed by

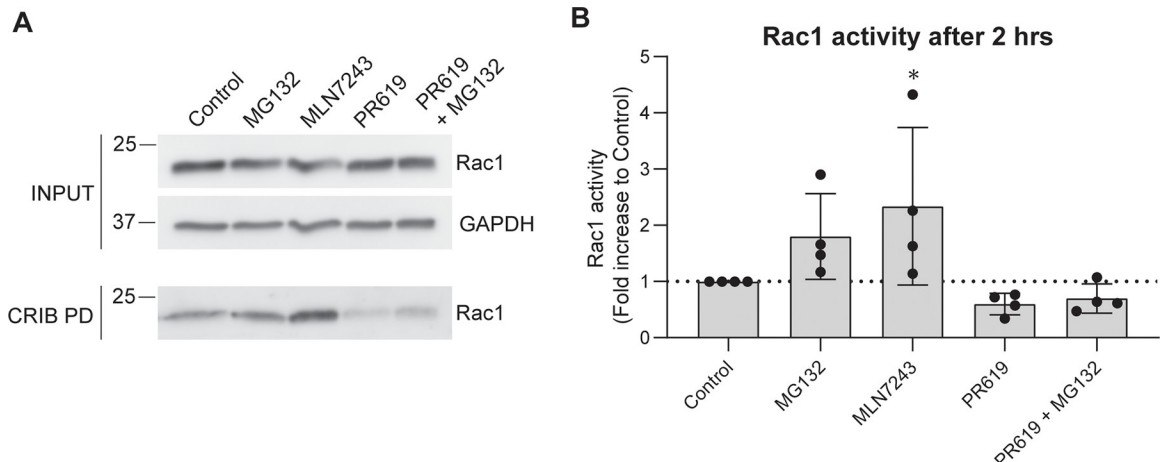

**Fig 1. Rac1 activity is regulated by the ubiquitin-proteasome system. A)** Representative western blot analysis of Rac1-GTP by CRIB pulldown (PD). HUVECs were treated for 2 hrs with the indicated inhibitors and pulldown of endogenous active Rac1 was performed as described in materials and methods. **B)** Quantification of Rac1 activity by densitometric analysis of western blot after CRIB pulldown (n = 4). * $p < 0.05$ compared to control in Holm-Sidak's post-hoc test of one-way ANOVA.

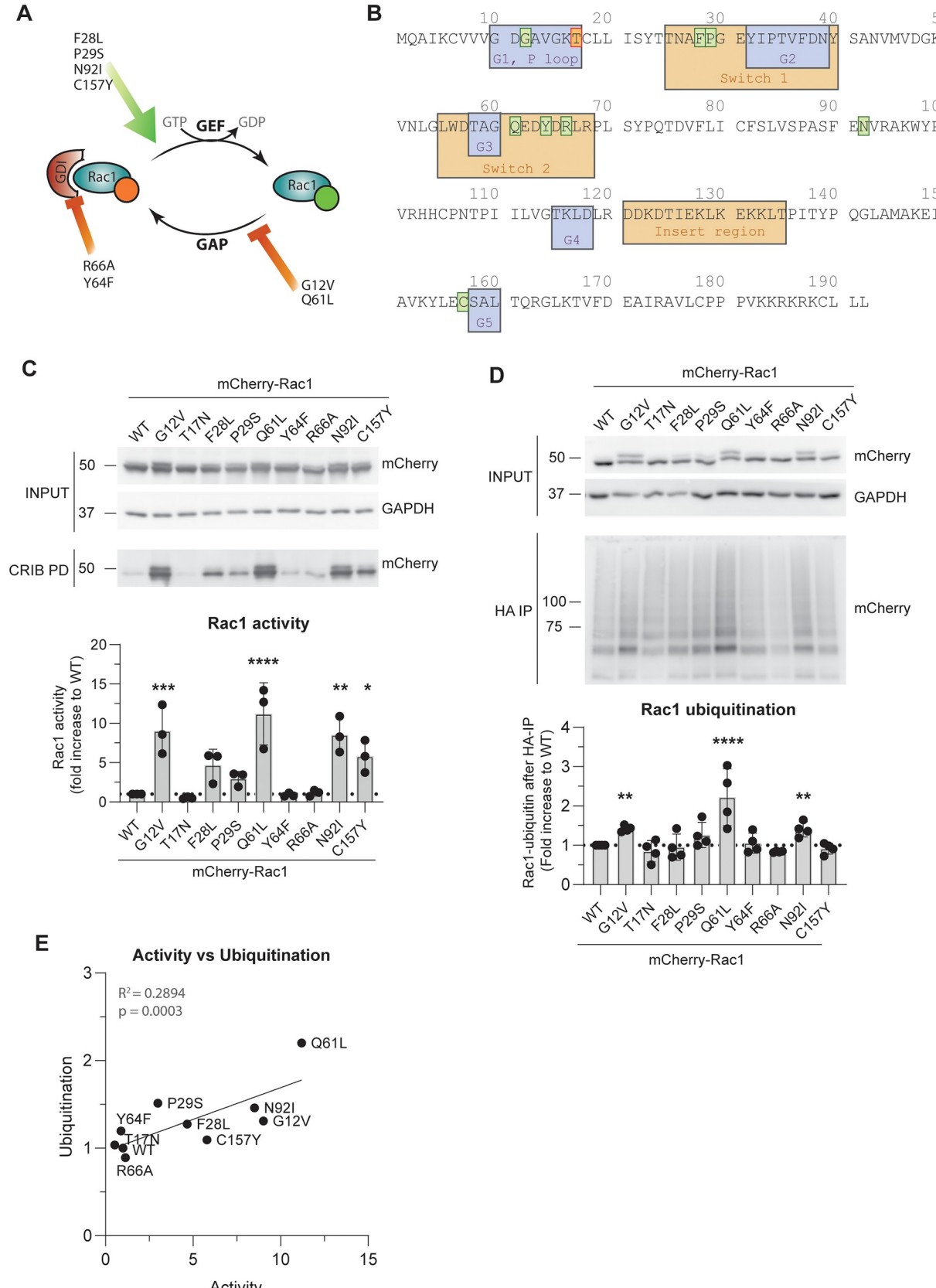

**Fig 2. Relationship between activity and ubiquitination of Rac1 in cells. A)** Predicted effects of the introduced Rac1 mutations on Rac1 cycling and activity. **B)** Amino acid sequence of Rac1 with indicated G1-G5, switch-1 and -2 and insert regions and highlighted amino acids mutated to activate (green) or inactivate (orange) Rac1. **C)** Representative Western blot and quantification of CRIB pulldown of mCherry-Rac1 and mutants after overexpression in HEK293T. * p<0.05, ** p<0.01, *** p<0.005 and **** p<0.001 in Holm-Sidak's post-hoc test of one-way ANOVA. **D)** Representative Western blot and quantification of ubiquitination assay in cells of mCherry-Rac1 and the indicated mutants after expression in HEK293T. ** p<0.01 and **** p<0.001 in Holm-Sidak's post-hoc test of one-way ANOVA. **E)** Correlation graph with linear regression of the ubiquitination versus activity levels of mCherry-Rac1 activating mutants.

Rac1 N92I with an 8.5-fold increase in activity. Rac1 C157Y showed a 5.7-fold increase, followed by Rac1 F28L with a 4.6-fold and Rac1 P29S with a 3.0-fold increase in activity. In contrast to previous observations by others [26, 30], we did not find an increase in activity for Rac1 Y64F and R66A. The Y64F mutation even decreased Rac1 activity to 0.8 compared to WT, close to the 0.7-fold change by the constitutive inactive mutant T17N. The increase in activity of Rac1 R66A was 1.1-fold compared to WT.

We next performed a ubiquitination assay for the same Rac1 mutants (Fig 2D) and found that, as shown before [19], Rac1 Q61L ubiquitination is significantly increased (2.2-fold) compared to Rac1 WT. In addition, we observed a significant 1.4-fold increased ubiquitination of Rac1 G12V and N92I compared to WT. Rac1 R66A was less ubiquitinated than WT, namely 0.8-fold, comparable to T17N. Total ubiquitination of Rac1 F28L, P29S, Y64F and C157Y was not significantly changed (1.0-. 1.3-, 1.1-, and 0.9- fold compared to WT, respectively) but correlation analysis of activity versus ubiquitination of all Rac1 activating mutants showed a significant, positive correlation, meaning that increased activity is correlated with increased ubiquitination ($R^2 = 0.2894$, p = 0.0003) (Fig 2E).

## Rac1 activating mutants differentially bind RhoGDI

It is widely accepted that Rac1-GDP binds to the GDI, whereas Rac1-GTP is not GDI bound, and can therefore associate with the plasma membrane and downstream effectors [5]. We investigated GDI binding of the mCherry-Rac1 activating mutants by performing a mCherry Co-IP with endogenous GDI protein in HEK293T cells (Fig 3A and 3B). We found that the

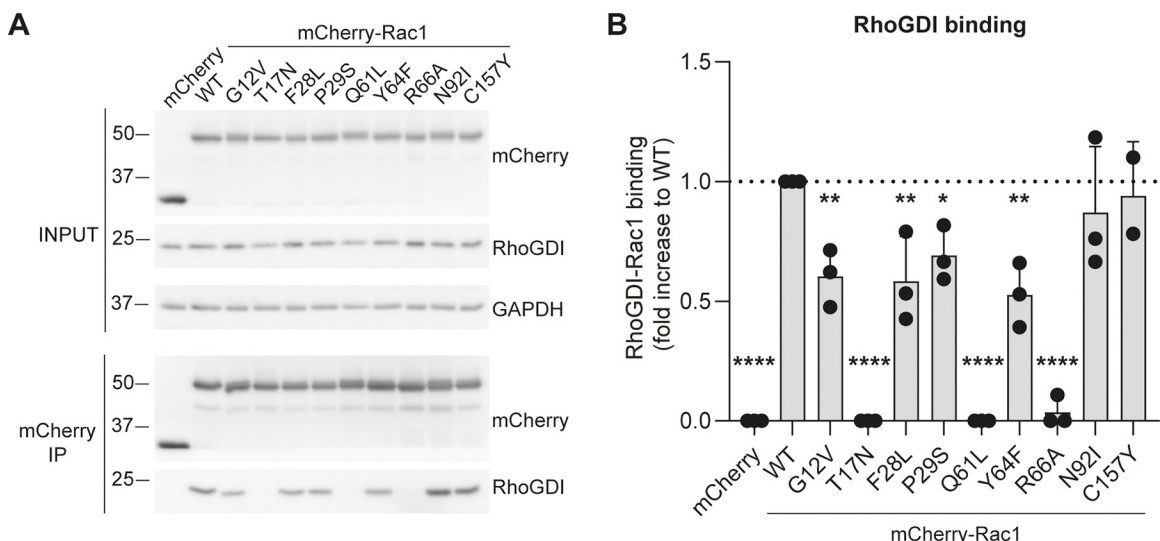

**Fig 3. GDI binding properties of Rac1 activating mutants in cells. A and B)** Representative Western blot (A) and quantification (B) of mCherry-Rac1 Co-IP with endogenous RhoGDI. MCherry-Rac1 was expressed in HEK293T cells, after which Co-IP for mCherry was performed as described in materials and methods. *p< 0.05, ** p< 0.01 and **** p<0.001 in compared to WT in Holm-Sidak's post-hoc test of one-way ANOVA (n = 3).

Rac1 mutants T17N, Q61L and R66A did not, or very little, bind the GDI, which is in accordance with previous research [6]. Rac1 N92I and C157Y bound the GDI to a similar extent as Rac1 WT, and the GDI-binding capacity of Rac1 G12V, F28L, P29S and Y64F was approximately 50% reduced compared to Rac1 WT (Fig 3A and 3B).

## All activating mutants except for F28L and R66A induce an active Rac1 phenotype in endothelial cells

Rac1 activity induces cell spreading and membrane ruffling, consequent to its induction of cortical actin polymerization. After testing the activity, ubiquitination and GDI binding of the Rac1 mutants in HEK293T cells, we investigated the morphological effects of these proteins upon expression in HUVECs seeded at low density to allow analysis of individual cell size and shape (Fig 4A). Between the mutants, we did not observe a significant difference in cell size, in

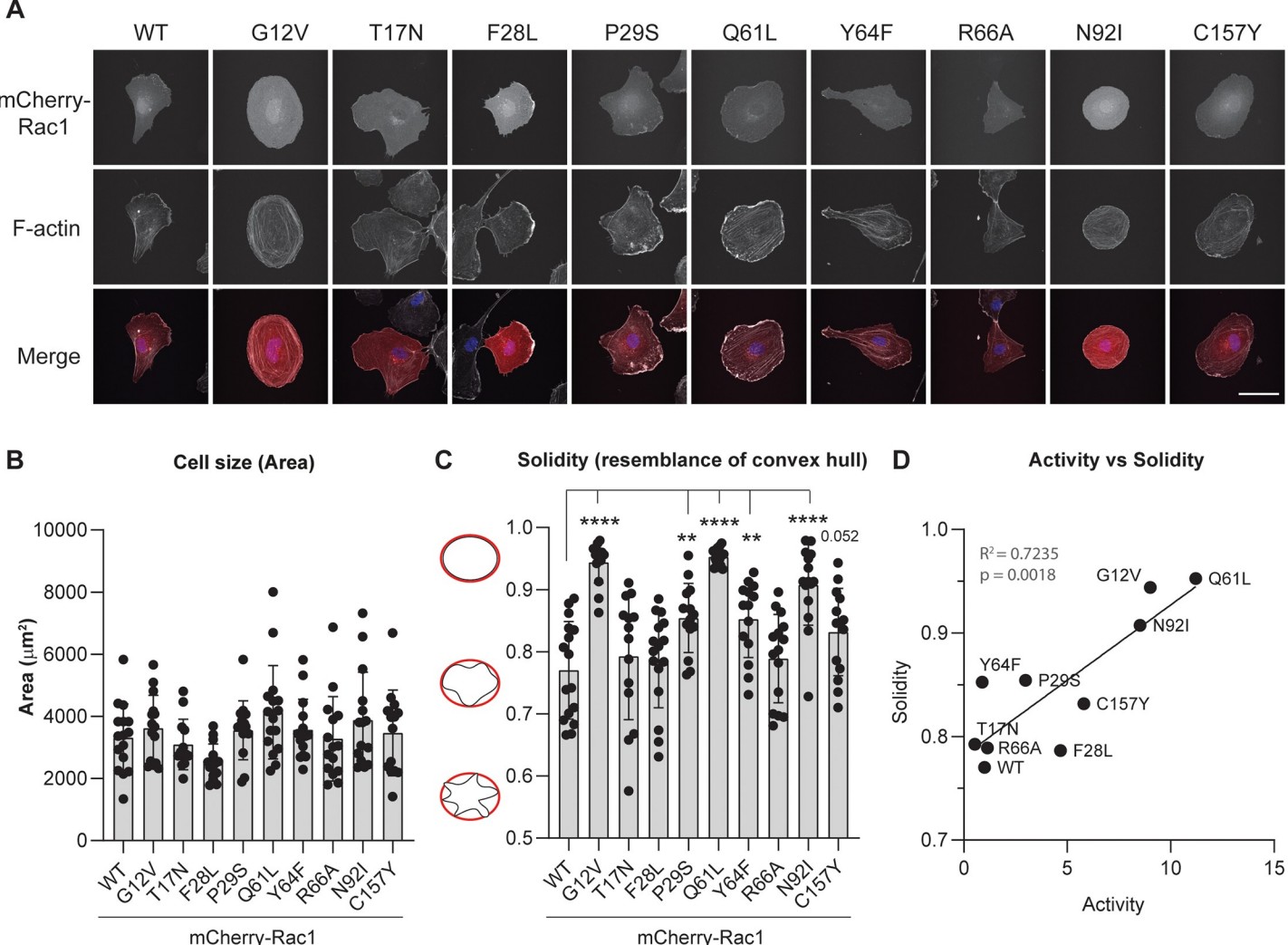

**Fig 4. Rac1 G12V, Q61L and N92I mutants induce comparable morphology change in HUVECs. A)** Immunofluorescent analysis of mCherry (red) and F-actin (gray) of HUVECs with transient overexpression of mCherry-Rac1 and mutants. Scale bar represents 50 μm. **B)** Quantification of cell size defined by surface area covered. **C)** Quantification of cell shape solidity of > 20 cells per sample. **p<0.01 and ****p<0.001 compared to WT in Holm-Sidak's post-hoc test of one-way ANOVA. **D)** Correlation graph with linear regression of the solidity versus activity levels of mCherry-Rac1 activating mutants.

part since the spread of the data is considerable (Fig 4B). As expected, the mutations G12V and Q61L induced clear lamellipodia formation in HUVECs and the concomitant cell morphology is markedly circular. The N92I mutation induced a phenotype similar to G12V and Q61L, as deduced from the quantification of cell shape solidity as description for cell deformability (Fig 4C) [31]. Interestingly, the P29S mutation showed a clear active Rac1 phenotype compared to Rac1 WT whereas the increase in solidity of Rac1 F28L was only moderate, although these mutations induced a similar increase in Rac1 activity, based on binding to the CRIB peptide (Fig 2C). Additionally, Rac1 Y64F activity was not increased in the CRIB pulldown compared to Rac1 WT but this mutant, be it only moderately, increased HUVEC shape solidity. Rac1 C157Y mutation also increases the solidity of the cell shape compared to Rac1 WT. As expected, the morphology of HUVECs expressing Rac1 T17N and Rac1 R66A was similar to cells expressing Rac1 WT. A correlation plot of the Rac1 activity measured by CRIB pulldown versus the HUVEC solidity shows a significantly positive correlation ($R^2$ = 0.7235, p = 0.0018), indicating that the measurement of Rac1 activity by CRIB pulldown is a valid indication for its biologically relevant activity.

## Mutation of lysine 16 of Rac1 impairs both activity and ubiquitination of constitutively active Rac1

Ubiquitination of proteins generally occurs on lysine residues that are exposed on the surface of the protein and thus accessible to the ubiquitin ligase [32]. Following our analysis of the correlation between Rac1 activity levels and Rac1 ubiquitination, we introduced lysine (K) to arginine (R) point mutations in myc-Rac1 Q61L based on published literature and on the tertiary structure of Rac1 (Fig 5A and 5B). The selected lysines are well conserved across species, except for K183 and K184, which deviate in D. rerio and D. melanogaster (S2 Fig). The K to R mutation prevents ubiquitination while retaining a positive charge to minimize the effect of the mutation on the tertiary structure of Rac1 (Fig 5A and 5B). As shown in Figs 2–4, Rac1 Q61L does not bind the GDI, is most active and is most ubiquitinated, which is why we choose this mutant as background for the K to R mutations.

We observed a significant increase in total ubiquitination of the myc-tagged Rac1 Q61L compared to WT as was shown previously [19] (Fig 5C). This increase was absent when lysines 16 or 147 in Rac1 Q61L were mutated (0.4- and 0.6- fold compared to Rac1 Q61L). Conversely, we found that the introduction of the K133R and K166R mutation increased total Rac1 Q61L ubiquitination levels 1.6- and 1.9-fold, respectively. The total ubiquitination of Rac1 Q61L/K183R and Rac1 Q61L/K184R was unchanged.

The activity of Rac1 Q61L/K16R and Q61L/K166R was significantly decreased compared to Rac1 Q61L (0.15- and 0.45-fold, respectively) while the activity of Rac1 Q61L/K184R was increased (1.2-fold) (Fig 5D) as measured by CRIB pulldown from transfected HEK293T cells. The activity of Rac1 Q61L/K133R, -K147R and -K183R was not significantly changed. The lack of Rac1 Q61L/K147R ubiquitination is in line with published literature, while the loss of ubiquitination of Rac1 Q61L/ K16R was not shown before in this setup. Rac1 K16, but not K147, has been proposed to be subject to TRAF6-mediated ubiquitination [17]. However, in our studies, Rac1 Q61L/K147R is not ubiquitinated, even with the K16 residue intact. Based on these results, CRIB pulldown data and structure model of K16R Rac1 mutant (S4 Fig), we suggest that the lack of K16R ubiquitination is most likely due to its impaired GTP-binding capacity.

Due to decreased activity of myc-Rac1 Q61L/K16R and Q61L/K166R mutants, we hypothesized that these mutants might bind to RhoGDI more efficiently. As previously shown by others, Rac1 Q61L does not interact with the GDI, whereas the wild type Rac1 protein does (Fig

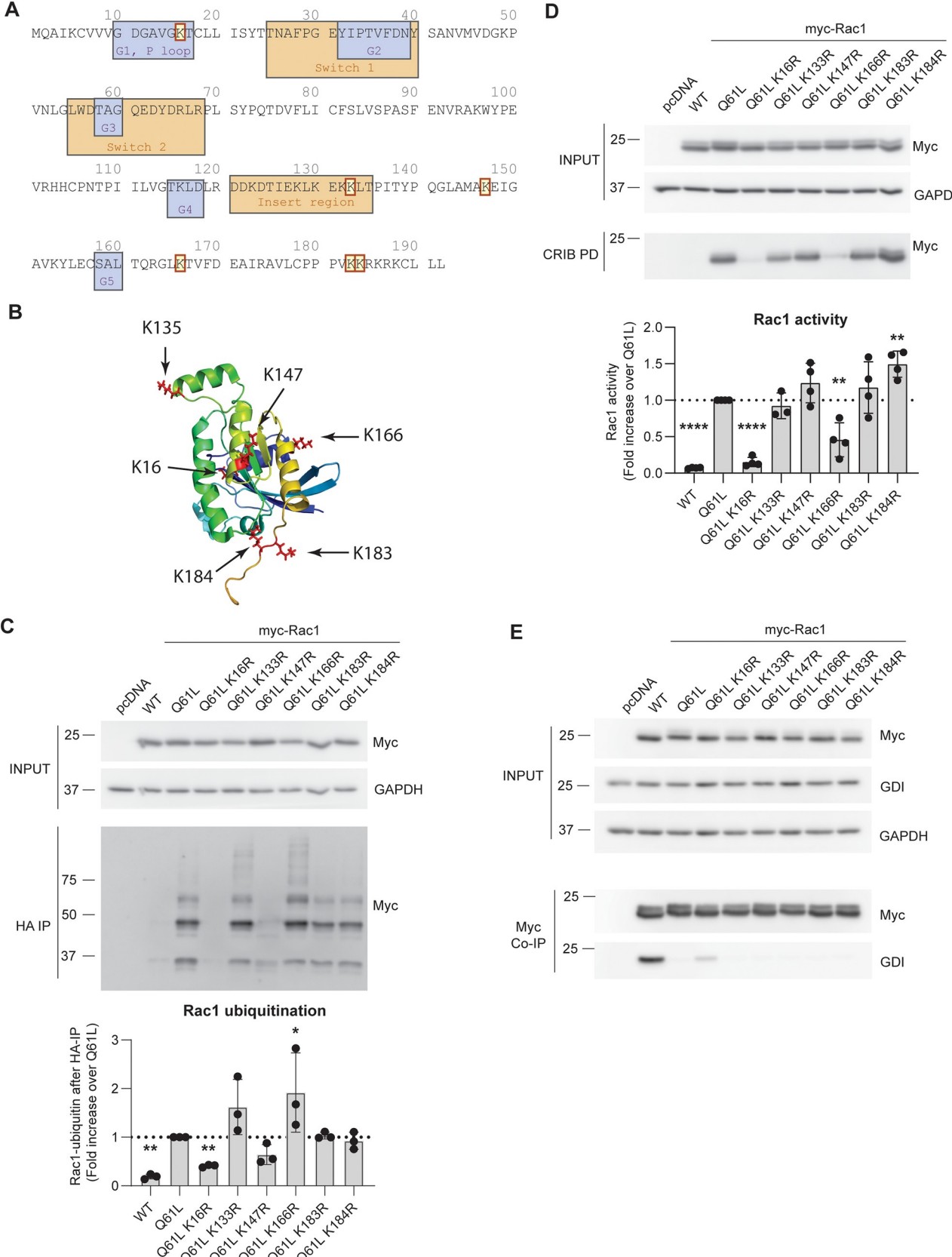

**Fig 5. Single K-to-R mutations alter Rac1 ubiquitination and activity. A)** Amino acid sequence of Rac1 with indicated G1-G5 regions, switch-1 and -2 and insert regions and (highlighted) lysine residues that were mutated to arginine. **B)** Tertiary structure of GTP-bound Rac1 with arrows pointing to the mutated lysines. Rac1 structure was obtained from the Rac1/PRK1 complex crystal structure 2RMK from RCSB protein data bank (rcsb.org) **C)** Representative Western blot and quantification of the ubiquitination assay. Myc-Rac1 and HA-Ubiquitin were expressed in HEK293T, after which denaturing HA immunoprecipitation was performed, followed by Western blot of the immunoprecipitates for myc-Rac1. * p<0.05 and ** p<0.01 after Holm-Sidak's post-hoc analysis of one-way ANOVA (n = 3). **D)** Representative Western blot and quantification of CRIB pulldown from overexpressed myc-Rac1 in HEK293T cells. ** p<0.01 and **** p<0.001 after Holm-Sidak's analysis of one-way ANOVA (n = 4). **E)** Representative Western blot of Co-IP showing endogenous GDI binding to immunoprecipitated myc-Rac1.

5E) [7]. Interestingly, the Q61L/K16R Rac1 mutant shows a modest recovery of GDI binding compared to Rac1 Q61L, but we found no change in GDI binding for Rac1 Q61L/K166R. This indicates that GDI binding does not correlate with a change in Rac1 activity of these K to R mutants.

To investigate morphological effects of the myc-Rac1 Q61L K to R mutants, we expressed the proteins in HUVECs (Fig 6A). Upon transient expression, myc-Rac1 Q61L clearly induces the expected phenotype including membrane ruffling and circumferential cell spreading, represented by the solidity parameter. The decrease in Rac1 activity in Rac1 Q61L/K16R, as compared to Rac1 Q61L (Fig 5D), was clearly reflected in the HUVEC phenotype. The decrease in activity of Rac1 Q61L/K166R had limited effects on HUVEC solidity, which was not statistically significant. We did not observe change in overall cell size for any of the Rac1 Q61L K-R mutants (Fig 6B). The solidity of ECs expressing Rac1 Q61L/K16R was significantly decreased compared to Rac1 Q61L (Fig 6C), while for all other mutants, there was no significant change in endothelial cell shape. We found a significant, positive correlation ($R^2$ = 0.8146, p = 0.0021) between the activity of the myc-Rac1 Q61L K-R mutants as measured by CRIB pulldown, and the solidity of HUVECs expressing the same mutants (Fig 6D). Together, these data show that, in addition to ubiquitination pattern, a selection of K-R mutations affect both Rac1 activity and its consequent morphological effects in HUVECs.

## Discussion and conclusion

We have generated comparative data on the relationship between Rac1 GDI binding, -activity and -ubiquitination using manipulation with inhibitors of the UPS, introducing either activating Rac1 mutations, or K to R mutations in constitutively active Rac1 Q61L. Throughout these studies, we used mCherry- or myc-tagged versions of Rac1. In line with published work, these tags by themselves did not induce changes in Rac1 activity, ubiquitination or localization [33, 34].

Firstly, we show that short-term manipulation of the UPS (2 hours) can change the activity of Rac1 in endothelial cells. Inhibition of DUBs decreased GTP-Rac1 levels, whereas the inhibition of the E1 ubiquitin ligase increased GTP-bound Rac1, both without changing total Rac1 levels. Although these effects were obtained with general inhibitors that do not target Rac1 specifically, these data indicate that ubiquitination, either directly or indirectly, plays an important role in the activation and inactivation cycle of Rac1.

Using several, be it confirmed or putative, activating mCherry-tagged Rac1 mutations G12V, F28L, P29S, Q61L, Y64F, R66A, N92I and C157Y, we directly compared the effect of different Rac1 activity levels on its ubiquitination and output. Hereby we were able to show a positive correlation between the level of Rac1 activity and its ubiquitination, and that this depends on the intrinsic Rac1 GTP binding only, rather than its binding to RhoGDI.

The Rac1 F28L and P29S mutants were extensively described previously, and induce a moderate increase in Rac1 activity. The mutations are located in the Switch 1 region of Rac1 and induce a fast-cycling phenotype, meaning that both GTP association and GDP dissociation are

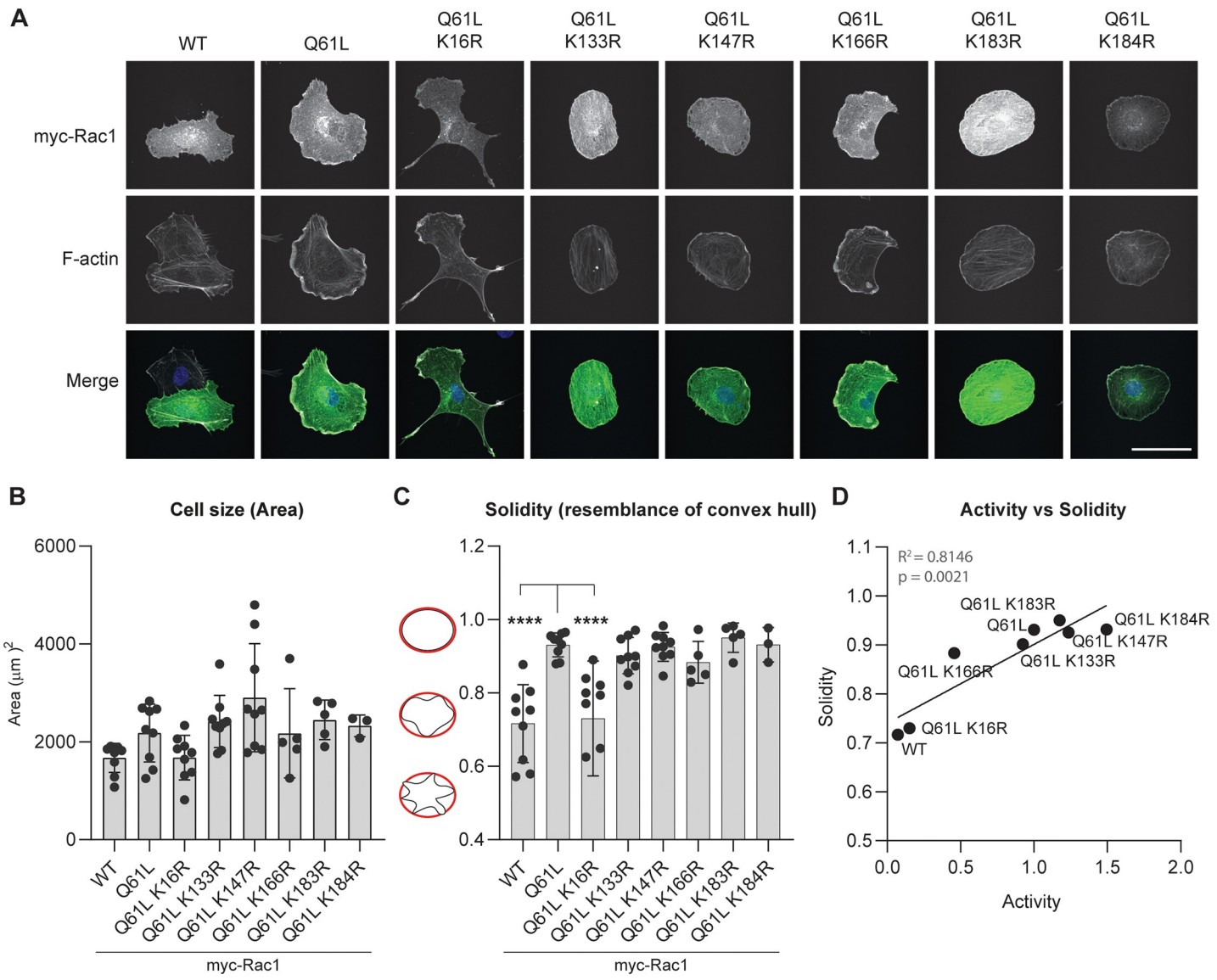

**Fig 6. Morphological effects of myc-Rac1 K-R mutants in HUVECs. A)** Immunofluorescent analysis of myc (green) and F-actin (gray) of HUVECs with transient overexpression of myc-Rac1 and mutants. Scale bar represents 50 μm. **B)** Quantification of cell size defined by surface area covered. **C)** Quantification of cell shape solidity of > 5 cells per sample. **** p<0.001 compared to Q61L in Holm-Sidak's post-hoc test of one-way ANOVA. **D)** Correlation graph with linear regression of the solidity versus activity levels of mCherry-Rac1 activating mutants.

accelerated [24, 25, 35–37]. Rac1 F28L and Rac1 P29S were included in our analyses to investigate moderate Rac1 activity, compared to the strong increase of activity by the G12V and Q61L mutations. In our assays, the activation of Rac1 by the G12V and Q61L mutations induced the greatest increase in Rac1 ubiquitination and moderate Rac1 activity in fast cycling mutants (F28L and P29S) lead to a moderate increase in Rac1 ubiquitination.

Remarkably, the Rac1 N92I mutation increased Rac1 activity and ubiquitination to a similar extent as the extensively studied Rac1 Q61L and G12V. The Rac1 N92I mutant still retains full GTPase activity, whereas the intrinsic GTPase activity of Rac1 Q61L and Rac1 G12V is impaired (0% and 5% GTP hydrolysis, respectively, over 10 minutes compared to 30% in Rac1 WT) [22]. It was suggested that the N92I mutation alters Rac1 activity by disrupting

intramolecular interaction with the D11 residue in the P-loop and thereby impairing GDP/GTP binding [20], although this mechanistic explanation still needs to be confirmed.

Rac1 C157Y was, just as Rac1 F28L, P29S and N92I, described as a fast-cycling mutant with modest increase in activity. This mutant however displays increased GTP dissociation in addition to GTP association and GDP dissociation, whereas GTP dissociation of Rac1 F28L, P29S and N92I is equal to Rac1 WT [20]. The C157Y mutation of Rac1 was also found in an individual with a developmental disorder and neurological disability. Although no activity differences were observed for this mutant in *in vitro* experiments in fibroblasts by Reijnders et al., the research from Kawazu et al. did show increased activity of the Rac1 C157Y mutant in HEK293T cells [20, 27]. In accordance with this latter study, our data show an increase in activity of Rac1 C157Y compared to Rac1 WT, albeit that its level of ubiquitination was not detectably increased, likely because the ubiquitination assay is less sensitive than the CRIB pulldown.

It has been generally accepted that only inactive Rac1 is bound to the GDI, which was why activation and degradation of Rac1 seemed to go seamlessly hand in hand, but a vast body of research has shown that the GDI regulates Rac1 activation in an active rather than passive manner [4, 5, 7, 38–42]. Based on the notion that loss of GDI binding induces activation and degradation of Rac1, we expected that Rac1 R66A, a mutant defective for GDI binding, would be either more active, or in any case more ubiquitinated and that we would see a decrease in total protein levels for this Rac1 mutant [26]. However, ubiquitination of Rac1 R66A is slightly decreased and there was no increase in Rac1 activity, although GDI binding was sufficiently inhibited (Fig 2C). Similarly, the Rac1 Y64F mutant was previously shown to decrease the Rac1-GDI interaction and increase Rac1 activity [30, 42]. Although we could confirm the decrease in GDI binding, we did not detect change in activity between Rac1 WT and Rac1 Y64F. Other two Rac1 activating mutants, F28L and P29S, both showed an approximate 50% decrease of GDI binding, but the GDI binding itself did not correlate with the activity or ubiquitination levels of these Rac1 mutants. Interestingly, the active N92I Rac1 mutant bound equally well to the GDI as WT Rac1, whereas the G12V mutation decreased GDI binding, and Rac1 Q61L showed no interaction with the GDI at all. The fact that Rac1 N92I binds equally well to the GDI as Rac1 WT makes this an interesting mutation to use in fundamental studies on the role of GDI binding and activation in Rac1 signaling. In conclusion, our data show that lack of GDI binding is not sufficient to induce activation of Rac1, that Rac1 activation does not exclude GDI binding and that GDI binding does not protect active Rac1 from ubiquitination.

Single lysine mutations have been used in many studies to identify site-specificity of ubiquitination by ubiquitin ligases [14, 16, 43], but we set out to also investigate the effect of these mutations on overall Rac1 ubiquitination and activity. Analysis of single lysine mutations in myc-Rac1 Q61L showed several opposing effects. The K16R mutation, as was described before [19], inhibits Rac1 activity even when the Q61L mutation is present and also fails to induce ruffling and the stereotypical circular cell shape that is associated with Rac1 activity in HUVECs. We have modelled the Rac1 structure with either K16 or R16 in interaction with GTP (S4 Fig) [44]. K16 of Rac1 interacts with the second and third phosphate group of GTP, with intramolecular distances being 2.65 and 2.75 Å, respectively. In the model of the Rac1 K16R mutant, the $N_\epsilon$ from R16 interacts with the second phosphate of GTP with a distance of 3.44 Å and $N_\zeta$ interacts with the third phosphate of GTP with a distance of 2.41 Å. This increase in distance between $N_\epsilon$ of R16 and the second phosphate of GTP could lead to strong reduction of the GTP binding in K16R mutant. Recent data from Acuner et al. shows that the GTP-K16 bond is lost in Rac1 Q61L with K16S and K16M mutations [45]. Although the K16R mutation was not investigated, it is plausible that this mutation also interferes with the GTP binding of Rac1. The possible necessity of K16 for GTP-binding in the P-loop of Rac1 was also

suggested by Vetter et al. [46]. This supposition is also in line with our data that the Rac1 Q61L/K147R mutant (in which K16 is intact) does not show ubiquitination while being active. This indicates that K147, in line with earlier findings [19, 47, 48], is strictly required for the ubiquitination of activated Rac1.

Compared to Rac1 Q61L, both Rac1 Q61L/K133R and Q61L/K166R displayed increased ubiquitination, although there was no change in activity for Rac1 Q61L/K133R and the activity of Rac1 Q61L/K166R was even decreased. We also did not observe clear changes in the activity of WT Rac1 with the same mutations (S3 Fig). K133 is located in the insert region and the side chains of both K133 and K166 are not involved in intramolecular interactions within Rac1, so mutation of these lysines may influence interactions with other proteins directly. In previous studies Rac1 K166R was found to be less degraded in MEFs and less ubiquitinated *in vitro* compared to Rac1 WT [43], but the effect of this mutation may be different in Rac1 Q61L/K166R versus Rac1 WT/K166R since we do not observe an increase in Rac1 Q61L/K166R protein levels.

Mutation of C-terminal K183 and K184 lysine residues did not reduce ubiquitination of Rac1 Q61L, although an approximate 50% increase in activity was observed for Rac1 Q61L/K184R. This increase in activity is remarkable given that it occurs in an already constitutively active Rac1 protein. Since lysine 183 and 184 are targeted for SUMOylation [48], the loss of SUMOylation at this location may explain the increase in activity [49]. Furthermore, orientation of the C-terminus of Rac1 with respect to the rest of the molecule is not resolved in the 3D structure. Therefore, it is difficult to predict which structural changes in Rac1 protein may be induced by the K183R and K184R mutations.

Taken together, this study compared a large number of (active) Rac1 mutants to further chart the relationship between Rac1-GDI binding, -activity and -ubiquitination. Although we cannot formally exclude that the pool of ubiquitinated Rac1 is distinct from the pool we found associated with the GDI, our data strongly suggest that the ubiquitination of Rac1 is closely linked to its level of activity and that GDI binding by activated Rac1 versions does not protect from increased ubiquitination. Further studies are required to define the localization of Rac1 ubiquitination and -degradation and to establish its role as an alternative Rac1-inactivating mechanism, next to GAP- stimulated GTP hydrolysis.

## Supporting information

**S1 Fig. Inhibitors used in this study do not directly affect Rac1-PAK interaction.** HUVECs were treated with indicated inhibitors for two hours before performing CRIB pulldown, or inhibitors were added in the lysis buffer only.
(PDF)

**S2 Fig. Rac1 protein alignment from multiple species, highlighting the conservation of the mutated lysines in these studies.**
(PDF)

**S3 Fig. K-R mutations do not significantly change activity of mCherry-Rac1 WT.**
mCherry-Rac1 and mutants thereof were ectopically expressed in HEK293T cells and CRIB pulldown was performed as described in the materials and methods. Representative Western Blot and analysis of n = 3 independent experiments.
(PDF)

**S4 Fig.** Predicted interaction of (A) Rac1 K16 and (B) Rac1 R16 showing differential interaction with the GTP molecule. Pymol software was used for the protein structure alignment of the GNP-loaded WT and K16R Rac1. The structure of the wild type Rac1 was obtained from

Protein Research Database (ID: 3th5) and the structure of the K16R mutant was obtained by modelling using the Phyre2 server (The Phyre2 web portal for protein modeling, prediction and analysis; [44]).
(PDF)

**S1 Raw images.**
(PDF)

## Acknowledgments

pcDNA3-HA-Ubiquitin was a kind gift of K. Husnjak, Goethe University Medical School, Frankfurt am Main, Germany.

## Author Contributions

**Conceptualization:** Jisca Majolée, Peter L. Hordijk, Igor Kovačević.

**Formal analysis:** Jisca Majolée, Fabienne Podieh.

**Funding acquisition:** Peter L. Hordijk.

**Investigation:** Jisca Majolée, Fabienne Podieh.

**Methodology:** Jisca Majolée, Fabienne Podieh, Peter L. Hordijk, Igor Kovačević.

**Supervision:** Peter L. Hordijk, Igor Kovačević.

**Visualization:** Jisca Majolée.

**Writing – original draft:** Jisca Majolée.

**Writing – review & editing:** Jisca Majolée, Peter L. Hordijk, Igor Kovačević.

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
