## [Decision Letter · Decision Letter 0]

25 Mar 2021

PONE-D-21-06139

The interplay of Rac1 activity, ubiquitination and GDI binding and its consequences for endothelial cell spreading.

PLOS ONE

Dear Dr. Kovacevic,

Thank you for submitting your manuscript to PLOS ONE. After careful consideration, we feel that it has merit but does not fully meet PLOS ONE’s publication criteria as it currently stands. Therefore, we invite you to submit a revised version of the manuscript that addresses the points raised during the review process.

The manuscript has been reviewed by two experts in the filed, who agreed that while the article is interesting conceptually, there area several concerning issues that would need to be addressed for this manuscript to be accepted for publication. Both reviewer reached similar conclusion and believe that the experiments performed are not sufficient to properly support the conclusions reached by the authors. Some key issues are the use of mCherry to tag Rac and also the effect of mutations such as the lysine mutants in the structure of Rac. There are also other technical concerns related to lack of proper controls and the effects of the inhibitors used that would need to be addressed if the authors decide to submit a revised version.

We look forward to receiving your revised manuscript.

Kind regards,

Rafael Garcia-Mata, PhD

Academic Editor

PLOS ONE

Journal Requirements:

Reviewers' comments:

Reviewer's Responses to Questions

**Comments to the Author**

1. Is the manuscript technically sound, and do the data support the conclusions?

Reviewer #1: Partly

Reviewer #2: Partly

2. Has the statistical analysis been performed appropriately and rigorously? 

Reviewer #1: Yes

Reviewer #2: Yes

3. Have the authors made all data underlying the findings in their manuscript fully available?

Reviewer #1: Yes

Reviewer #2: Yes

4. Is the manuscript presented in an intelligible fashion and written in standard English?

Reviewer #1: Yes

Reviewer #2: Yes

5. Review Comments to the Author

Reviewer #1: In this manuscript, Majolée et al. explore relationships between Rac1 activity, GDI binding and ubiquitination. There is a lot of good data of interest to the cell biology community in this manuscript around the effects of several Rac1 mutants on Rac1 localization and function in cell lines. However, the overall conclusions in terms of effects on Rac1 ubiquitinylation, GDI interactions and activity are not convincing or complete, as detailed below.

Major

1. In Fig. 1, the authors incubate HUVECs for 2 h with ubiquitin and proteasome associated inhibitors and then follow Rac1 activity by PAK-CRIB pull-down and Rac1 Western blot. The authors conclude that there is some role of for ubiquitin and proteasome in regulating Rac1 activity. The results are interesting, but slightly difficult to interpret without the inclusion of any controls. For example, it is not clear if the compounds actually affect Rac1 activity, or simply an interaction between Rac1 and PAK. Does the compound have any effect on CRIB pulldown of pure GDP or GTPγS-loaded Rac1?

2. It is not clear whether HUVECs in their basal state are most appropriate for these kinds of studies – or if some stimulated state that activates Rac1 would be more informative (i.e., thrombin stimulation).

2. PAK-CRIB assays can have high variation, and it is best to include positive and negative controls to give some insight to min and max values – typically by adding excess GDP or GTPγS to the sample to prevent or enhance Rac1-PAK interactions.

3. Do the inhibitor treatments in Fig. 1 also enhance/prevent CRIB pulldown of Cdc42? Do inhibitors have an effect on PAK autophosphorylation as a marker of downstream Rac1-GTP activity, rather than just an interaction with recombinant CRIB domain?

4. In Fig. 2D, it is not clear if site-specific mutants of Rac1 that affect Rac1 activity are altering Rac1 ubiquitinylation – or, if Rac1 activity itself globally regulates ubiquitinylation. Did the authors measure ubiquitinylation of any other specific proteins or total cell lysates in these samples? For instance, the Q61L lane seems to have more background signal in general. Overall, it is very concerning that the authors never examine total protein ubiquitinylation in their samples (anti-ub Western blot), as the results could just be an effect of protein overexpression on homeostasis.

5. How many different RhoGDIs are expressed by HEK293T cells and which one is examined in Fig. 3? The authors are likely following RhoGDI�/RhoGDI1/ARHGDIA – but RhoGDI2/ARHGDIB may be more relevant to Rac1 interactions and localization.

6. A main conclusion of the manuscript appears to be “that the lack of K16 ubiquitination is primarily due to a loss of its GTP-binding capacity”. However, the data supporting this conclusion are not very convincing. It seems more likely that mutations (even K->R) within the Rac1 GTP binding domain severely alter something about Rac1 structure/function in a manner limiting any function around Rac1. The authors do provide some control information here in Supp. Fig. 2 demonstrating that K16R on its own does not affect Rac1-CRIB interactions, but it is difficult to distinguish “zero” levels from background without an mCherry alone or even blank pulldown control.

7. Controls for uniform transfection and consistent levels of exogenous protein expression are not clear.

Minor

1.The Abstract includes too much background information, is too vague and should include more specific summaries of Results and Conclusions.

2. In general, many parts of the manuscript are too wordy and focused on Background information. There is no need to repeat Background details again and again in Abstract, Intro, Results, etc. There is so much editorial language in the Results section, that it is difficult to

3. The use of the term “in vivo” in reference to Fig. 2 is not ideal - suggests an effect in an animal model or human subject.

4. Fig. 3 is informative – but perhaps would be best kept as a Supplementary figure confirming previous results – or as part of a larger figure. For instance, it might make the presentation of the data better if the authors aligned the graph in Fig. 3B above similar graphs in Fig. 4BC to show how GDI interactions and Rac1 activity relate.

5. Fig. 5B is very helpful. It would be nice to have a related (perhaps Supplemental) panel (or reference citation) aligning Rac1 protein sequences from multiple species to show lysine conservation.

6. Is there a reason for difference in Rac1 staining of KR mutants in Fig. 6? This appears to be independent of effects on Rac1 activity and could be of interest.

Reviewer #2: In the manuscript entitled “The interplay of Rac1 activity, ubiquitination and GDI binding and its consequences for endothelial cell spreading”, Majolee et al investigate the relationship between Rac1 activity, ubiquitination and binding to RhoGDI using a range of Rac1 mutants including activating, inhibitory or fast-cycling mutations. The authors positively correlate the activity and ubiquitination of the mcherry-Rac1 mutants They also assess endothelial cell spreading upon expression of the various mutants and positively correlate the solidity of EC shape (cell spreading) with Rac1 activity (as measured by CRIB pull-down assay in all cases). They next set out to assess the effect of ubiquitination on Rac1 activity and RhoGDI binding by individually mutating a number of lysine residues in a Q61L Rac1 mutant background. The K16R mutation blocks both activation and ubiquitination of Rac1.

Overall, this manuscript is interesting as the authors set out to investigate the relationship between Rac1 activity, ubiquitination and RhoGDI binding. It is clearly written and scientifically sound regarding the experiments carried out and the conclusions drawn by the authors. I think the authors satisfactorily correlate endothelial cell spreading with Rac1 activity, so I won’t comment these experiments any further. However, I think the conclusions that can be drawn from this set of experiments and the underlying strategy are somehow limited because it is difficult, solely based on the experiments provided, to determine which effects are dependent on the ubiquitination and on the mutations on Rac1 structure. Therefore, I wouldn’t recommend it for publication unless additional experiments would support and back up the authors conclusions. It is definitely conceptually appealing but falls short of addressing a critical issue under this form in my opinion.

Major points:

My understanding of the regulation of Rac1 by ubiquitination and degradation was that this was a mean to terminate Rac1 activation in case of sustained or irreversible Rac1 activation. For instance, following CNF1 treatment, Rac1 gets irreversibly modified and constitutively activated which triggers its proteasomal degradation. Here, I am struggling to understand the authors conclusions and how they include ubiquitination in the GTPase cycle.

In the first part of the manuscript, they assess the correlation between activity, ubiquitination and GDI binding. I think it is fair to conclude from their experiments that activity and ubiquitination and dissociated from GDI binding with the mutants evaluated here, which incidentally does not necessarily mean that endogenous Rac1 behaves similarly. I think it would be worth noting that accordingly to the model discussed above, non-cycling active mutants (Q61L, G12V) are more ubiquitinated than activated fast-cycling mutants (P29S, F28L).

One major criticism here that is not addressed or mentioned by the authors would be that they use mcherry-Rac1 constructs. This construct is most likely properly prenylated, but the potential caveat is that mCherry creates a massive steric hindrance on Rac1 that may affect its behavior (as compared to endogenous Rac1 or myc-tagged Rac1 used afterwards). This has been mentioned by others and justified the use of alternative tagging strategies in other studies (Golding et al., 2019). This seems quite critical to me to at least address this point since the focus of this manuscript is to assess the cycling of Rac1.

I am more skeptical with the second half of the manuscript. The authors set out to investigate the effect of ubiquitination on the activity of Rac1 by using the classical active non-cycling Q61L mutant and individually mutating a number of selected Lysine residues. As previously, the authors compare activation, ubiquitination and RhoGDI binding. The Q61L K16R mutant stands out by its reduced activity which correlates with its reduced ubiquitination. This is where I am mostly conceptually puzzled:

Regarding ubiquitination, it looks like mutating this single residue completely blocks any ubiquitination of Rac1, does that mean this is indeed the only ubiquitinated residue on Rac1 or that it primes Rac1 for further ubiquitination on other Lysines?

I am not sure to understand the mechanistic effect of this mutation. My understanding is that the Q61L mutation favors GTP loading and prevents any hydrolytic activity hence increased activity. Does it mean that the K16R mutation prevents GTP loading of Rac Q61L? Therefore, this would mean that GTP loading would be ubiquitination-dependent which is quite controversial regarding what is known from CNF1 studies for instance.

Alternatively, as mentioned by the authors in the discussion, another explanation could simply be that this K16R mutation somehow disrupts Rac structure/function/ GTP binding… This could also hold true with any of the other Lysine mutants and makes it difficult to interpret the data and draw conclusions solely based on these experiments. I fear that unless the authors provide additional data to establish that these mutants can be used safely, it is difficult to confidently conclude anything from those experiments.

Minor point

The introduction is clear and states the rationale for this study, but I had the feeling that a number of (appropriate) statements would benefit from citing the articles demonstrating the statement such as lines 38, 41, and 49 for instance.

6. PLOS authors have the option to publish the peer review history of their article (what does this mean?). If published, this will include your full peer review and any attached files.

Reviewer #1: No

Reviewer #2: No

---

## [Author Response · Author response to Decision Letter 0]

28 May 2021

All responses to specific comments raised by reviewers are in the uploaded file "Response to Reviewers".

---

## [Decision Letter · Decision Letter 1]

25 Jun 2021

The interplay of Rac1 activity, ubiquitination and GDI binding and its consequences for endothelial cell spreading.

PONE-D-21-06139R1

Dear Dr. Kovacevic,

We’re pleased to inform you that your manuscript has been judged scientifically suitable for publication and will be formally accepted for publication once it meets all outstanding technical requirements. I would like to commend the team for their responsiveness to the reviewer's comments, which in my opinion, has strengthened the manuscript significantly.

Kind regards,

Rafael Garcia-Mata, PhD

Academic Editor

PLOS ONE

Reviewers' comments:

Reviewer's Responses to Questions

**Comments to the Author**

1. If the authors have adequately addressed your comments raised in a previous round of review and you feel that this manuscript is now acceptable for publication, you may indicate that here to bypass the “Comments to the Author” section, enter your conflict of interest statement in the “Confidential to Editor” section, and submit your "Accept" recommendation.

Reviewer #1: All comments have been addressed

Reviewer #2: All comments have been addressed

2. Is the manuscript technically sound, and do the data support the conclusions?

Reviewer #1: Yes

Reviewer #2: Yes

3. Has the statistical analysis been performed appropriately and rigorously? 

Reviewer #1: Yes

Reviewer #2: Yes

4. Have the authors made all data underlying the findings in their manuscript fully available?

Reviewer #1: Yes

Reviewer #2: Yes

5. Is the manuscript presented in an intelligible fashion and written in standard English?

Reviewer #1: Yes

Reviewer #2: Yes

6. Review Comments to the Author

Reviewer #1: The authors have more than adequately addressed the concerns of the reviewer. The extensive responses and rigor are appreciated.

Reviewer #2: The discussion with the authors was very fruitful and although I do not necessarily agree with all of their points, I think the data they present here is worth publishing in the interest of the field. As long as their data and other published work are presented and referred to fairly, I think this manuscript could stem the discussion and the reader will be able to draw its own conclusions and agree/disagree with the authors.

7. PLOS authors have the option to publish the peer review history of their article (what does this mean?). If published, this will include your full peer review and any attached files.

Reviewer #1: No

Reviewer #2: No

---

## [Editor Report · Acceptance letter]

2 Jul 2021

PONE-D-21-06139R1 

The interplay of Rac1 activity, ubiquitination and GDI binding and its consequences for endothelial cell spreading. 

Dear Dr. Kovačević:

I'm pleased to inform you that your manuscript has been deemed suitable for publication in PLOS ONE. Congratulations! Your manuscript is now with our production department. 

Kind regards, 

on behalf of

Dr. Rafael Garcia-Mata 

Academic Editor

PLOS ONE